# Endocrine-Disrupting Chemicals and Persistent Organic Pollutants in Infant Formulas and Baby Food: Legislation and Risk Assessments

**DOI:** 10.3390/foods12081697

**Published:** 2023-04-19

**Authors:** Eleftheria Hatzidaki, Marina Pagkalou, Ioanna Katsikantami, Elena Vakonaki, Matthaios Kavvalakis, Aristidis M. Tsatsakis, Manolis N. Tzatzarakis

**Affiliations:** 1Department of Neonatology & NICU, University Hospital of Heraklion, 71003 Heraklion, Crete, Greece; 2Medical School, University of Crete, 71003 Heraklion, Crete, Greece; 3Laboratory of Toxicology Science and Research, Medical School, University of Crete, 71003 Heraklion, Crete, Greece

**Keywords:** baby food, infant formulas, endocrine disruptors, legislation, persistent organic pollutants, pesticides, plasticizers, risk assessment, exposure limits

## Abstract

Human milk is the healthiest option for newborns, although, under specific circumstances, infant formula is a precious alternative for feeding the baby. Except for the nutritional content, infant formulas and baby food must be pollutant-free. Thus, their composition is controlled by continuous monitoring and regulated by establishing upper limits and guideline values for safe exposure. Legislation differs worldwide, although there are standard policies and strategies for protecting vulnerable infants. This work presents current regulations and directives for restricting endocrine-disrupting chemicals and persistent organic pollutants in infant formulas. Risk assessment studies, which are limited, are necessary to depict exposure variations and assess the health risks for infants from dietary exposure to pollutants.

## Contents

1.Introduction2.Pesticides
2.1.Current legislation and policies for pesticides2.2.Infant risk assessments for pesticides

3.Phthalates
3.1.Current legislation and policies for phthalates3.2.Infant risk assessments for phthalates

4.Parabens
4.1.Current legislation and policies for parabens4.2.Infant risk assessments for parabens

5.Bisphenols
5.1.Current legislation and policies for bisphenols5.2.Infant risk assessments for bisphenols

6.Dioxins, furans, and polychlorinated biphenyls
6.1.Current legislation and policies for dioxins, furans, and polychlorinated biphenyls6.2.Infant risk assessments for dioxins, furans, and polychlorinated biphenyls

7.Conclusions

## 1. Introduction

Human milk is the healthiest option for newborns because it provides babies with antibodies not found in formulas, is more easily digested, and infants can absorb fat and nutrients better. However, when infant formula cannot be avoided, it becomes a precious alternative for feeding the baby. The global infant formula market is expected to grow annually at 9.35% from 2017 to 2025, driven by the increasing working women population across various geographic regions worldwide. The manufacturers are focusing on enhancing the nutritional content of infant formula to substitute breastfeeding [1]. Except for the nutritional content, infant formulas and baby food must be free from chemical pollutants and contaminants. Thus, their composition is controlled by continuously monitoring the chemical concentrations and regulated by establishing upper limits and guideline values for safe exposure. Legislation differs worldwide, although there are standard policies and strategies for protecting vulnerable infants. 

Environmental pollution could not be natural as the chemicals are part of modern life. Organic pollutants have been found to contaminate human milk, infant formula, and infant food [2,3,4]. Infants and babies are dietary exposed or exposed by crawling or mouthing plastic things. The absorption, distribution in systems, and metabolism of xenobiotics in infants differ from adults due to differences in the circulatory system and the enzymes involved in these processes. The excretion of substances occurs to a lesser extent due to an immature metabolism and reduced renal function, which is increased during the first year of life. The role of hormones in cell differentiation and organ formation during prenatal development is most important [5]. The immune system and lungs develop from birth until the child is ten years old. The central nervous and reproductive systems fully develop during adulthood, indicating that, except for prenatal exposure, exposure during childhood and adulthood is also important [5]. The alterations in sex hormones during the fetal period and infancy affect the normal development of the reproductive system, resulting in chronic health problems. Thyroid function is also susceptible to hormone mimics as they compete with normal thyroid hormones for binding to transport proteins and TH-receptors [6,7,8,9,10]. Groups of endocrine or potential endocrine disruptors, their applications, and associated health effects are presented in Table 1. 

The list of chemicals that have been detected in infant milk and baby foods includes flame retardants, organochlorine pesticides (OCPs), polychlorinated biphenyls (PCBs), dioxins, and furans that are either produced industrially or by-products from industrial processes and combustion [4]. They are categorized as Persistent Organic Pollutants (POPs) that are dangerous and lipophilic with resistance to decomposition processes in the environment and organisms. They are transported by wind and water to other areas, and their accumulation in the food chain has resulted in their exposure to living organisms, even though some compounds have been banned. The Stockholm Convention (2001) listed 12 POPs, nine OCPs (aldrin, chlordane, dichlorodiphenyltrichloroethane-DDT, dieldrin, endrin, heptachlor, mirex, toxaphene, and hexachlorobenzene-HCB), PCBs, dioxins, and furans to be subjected to restrictions to protect human health and the environment [14]. 

Except for POPs, there are organic compounds that raise concerns due to their human toxicity, and they are not accumulating in the food chain and living organisms. Such chemicals are bisphenol A and its analogs, F and S, phthalate esters, and parabens that are used as additives and plasticizers in consumer products (cosmetics, clothes, food packaging, canned food, personal care products, and baby products), resulting in the everyday exposure of the general population and vulnerable groups to these chemicals [9,16,17,18]. Additionally, they have been characterized as endocrine disruptors or chemicals with potential endocrine action. In addition to their direct effects on hormonal function, they negatively impact the nervous, respiratory, and reproductive systems. Moreover, they seem to play a role in metabolic syndromes (obesity, type 2 diabetes) and cancer [9,19,20,21,22]. 

This review presents infant formula strategies and regulations for controlling specific pollutants (pesticides, phthalates, bisphenols, parabens, dioxins, furans, and PCBs). Recent studies on risk assessments that associate infants’ dietary exposure with potential health risks are also discussed to examine whether the scientific results agree with the legal requirements.

## 2. Pesticides

The group of pesticides includes insecticides, herbicides, and fungicides that are widely used in agriculture and the production of fruits, vegetables, and cereals for their protection against microorganisms or insects. Humans are exposed to pesticides either environmentally by domestic use or consumption of contaminated food or occupationally by farmers and workers in the industry [14,23]. An important source of an infant’s dietary pesticide exposure is drinking water that is used for the preparation of meals. The water used for industrial baby food production undergoes a unique cleaning process. Two large organic groups of great concern and toxicity are organophosphates and organochlorines. This group of pesticides includes a variety of chemical compounds, also inorganic, but it is impossible to study all of them.

Organophosphates (OPs) include malathion, parathion, phorate, diazinon, chlorpyrifos, tribufos, omethoate, disulfoton, and others. OPs in the human body are metabolized to less toxic compounds and excreted as specific or non-specific metabolites (dialkyl phosphate metabolites—DAPs). Infants can be exposed to OPs via infant formulas [3,24,25]. Although data in the literature are contradictory [26], there are indications for associations between prenatal or early life exposure and effects on neurodevelopment [27], poor intellectual development in 7-year-old children [28], respiratory problems [29], hypospadias in offspring [30], reduced birth weight [31,32], and even childhood leukemia [33]. 

Organochlorine pesticides (OCPs) include compounds that have been banned or subjected to strict regulation due to their high bioaccumulation and toxicity to humans, animals, and aquatic life. OCPs include hexachlorobenzene (HCB), dichloro diphenyl trichloroethane or 1,1,1-trichloro-2,2-bis (4-chlorophenyl) ethane (DDT), lindane or gamma-hexachlorocyclohexane (γ-HCH), dieldrin, heptachlor, and chlordane. Since the 1970s (DDTs), compounds that have been banned are still detected in human matrices and the whole food chain [34,35,36,37]. In the human body, DDT is metabolized to dichlorodiphenyldichloroethane (DDD), which is more water soluble and less toxic, and dichloro diphenyl trichloro ethylene (DDE), which accumulates in adipose tissue and poses a threat to health due to its long half-life [14]. The exception to the high lipophilic nature of OCPs is α-Hexachlorocyclohexane (α-HCH), which is water soluble and has a low bioaccumulation rate. OCPs are mostly undetectable in baby products, such as infant formula, due to the careful selection and extensive processing of raw materials [38]. 

Chlorate, an inorganic compound, will be specially mentioned in the present study, given its recent reevaluation by the European Commission regarding infant exposure. It is used as an herbicide but is currently banned in the European Union. The toxicity of perchlorate and chlorate anions lies in their ability to inhibit iodine uptake by the thyroid gland and disturb the production of thyroid hormones [39,40]. Although this exposure can be considered low because the concentration is in the ppb levels depending on the region, early life exposure to OCPs, given their high bioaccumulation rate, is worrying for the subsequent course of the body’s health [41].

### 2.1. Current Legislation and Policies for Pesticides

In Europe, Directive 2006/125/EC [42] set the guidelines for pesticide residues in cereal-based and baby foods that should not exceed the amount of 0.01 mg/kg food. However, for some pesticides, this amount exceeds the maximum permissible daily intake for infants and young children, which is lower than 0.0005 mg/kg body weight. These pesticides, such as aldrin and dieldrin, may already be prohibited, but due to their slow removal from the environment, they may still be detected in cereal-based and baby foods. Except for aldrin and dieldrin, disulfoton (also its sulfoxide and sulfone), fensulfothion (summed analogs), fentin (triphenyltin cation), haloxyfop (salts and esters), heptachlor (including the trans-heptachlor epoxide), hexachlorobenzene, nitrofen, omethoate, terbufos (also its sulfoxide and sulfone), and endrin are substances that are prohibited for use in agriculture that is intended for cereal-based and baby foods. 

According to Regulation (EU), No 609/2013 [43], the maximum residue levels of pesticides in food for infants and young children (1 to 3 years old) should be set at the lowest achievable level. According to this regulation, banning or limiting the use of pesticides would not guarantee that food for infants and children is free from these compounds because the environment is extensively contaminated. However, applying good agricultural practices and limiting environmental contamination could reduce food pollution. 

The Regulation (EU) 2016/127 [44] on food for specific groups set new labeling rules on infant formulas and was applied on 22 February 2020. 

The Commission Implementing Regulation (EU) 2020/585 of 27 April 2020 [45] concerns a coordinated multiannual control program of the Union (2021–2023) to ensure compliance with maximum residue levels of pesticides and to assess the consumer exposure to pesticide residues in and on the food of plant and animal origin. According to the relevant recommendation from the European Food Safety Authority (EFSA), foods intended for infants and young children will also be evaluated. 

Specifically for chlorate and perchlorate, after years of evaluation, the European Commission set new maximum residue levels for the compounds in foodstuffs in 2020 due to the increased use of disinfectants during the COVID-19 pandemic. Additionally, based on the data on the risk to public health in 2014, the mean dietary exposures to chlorate in European countries exceeded the Tolerable Daily Intake (TDI) in infants and young children (EFSA TDI for chlorate 3 μg/kg body weight/day and perchlorate 0.7 μg/kg body weight/day). Therefore, the Commission Regulations 2020/685 [46] and 2020/749 [47] clearly defined the maximum permitted levels of perchlorate, especially for infant and young children formulas and foods, baby food and processed cereal-based food at 0.01 mg/kg, 0.02 mg/kg and 0.01 mg/kg, respectively. 

In the United States, the EPA (Environmental Protection Agency) is responsible for the establishment of pesticide residue limits (tolerances) for foods, and the FDA (Food and Drug Administration) for the enforcement of those limits in imported and domestic food commodities and monitoring in the market. The limits have been set after many studies based on the toxicity of the substance and its decomposition products, the necessary amount and frequency, the possible routes of exposure, and the amount of residue on the food after application. These limits are stated through the Code of Federal Regulations and are renewed annually. Under the Food Quality Protection Act (FQPA), the EPA evaluates new and existing pesticides to ensure that they are used at safe levels for infants, children, and adults and reevaluates each pesticide every 15 years. Additional safety factors are established for the safety of pesticide use on food to consider the uncertainty in data relative to children. The information on food consumption by infants and children and pesticide residues in food are combined to perform dietary risk assessments to ensure that all guideline values are safe. Since the FQPA was passed in 1996, domestic uses and uses on the food of many pesticides were banned or restricted (such as many carbamates and most organophosphates) because they were considered dangerous for younger ages, the permissible residue limits for many foods intended for children were lowered, and children’s exposure to pesticides was even decreased by 70% [48,49]. The agency concluded that it was necessary to revoke all tolerances (maximum residue limits) for chlorpyrifos because registered uses of the compound result in high exposures that exceed safe levels, posing an increased health risk for the population and vulnerable groups. This final rule has been effective since 29 October 2021, and the tolerances expired on 28 February 2022.

The Pest Management Regulatory Agency (PMRA), an organization responsible for pesticide control, was established in Canada in 1995. When setting restrictions and prohibitions on pesticides, particular emphasis is placed on the young population, from the embryonic stage to childhood, as they are at greater risk. This is due to biological reasons and additional exposure that may result from powdered milk, breast milk, and increased consumption of fruits and vegetables. In 2005, the PMRA compiled a list of Pest Control Product Formulants and Contaminants of Health or Environmental Concern, updated annually. According to the 2020 directive, several substances, including coal-tar creosote, dimethyl formamide, dioctyl phthalate, isophorone, rhodamine B, adipic acid, bis (2-ethylhexyl) ester, and hydroquinone were removed from products suspected of health or environmental problems [50,51]. 

### 2.2. Infant Dietary Risk Assessments for Pesticides

A French study combined pesticide contamination of baby foods and common foods and consumption data since 2005 to estimate the probability of exposure exceeding the toxicological reference value (TRV) for infants and young children below three years old [52]. Pesticides were detected in 67% of the samples, although no exceedance of MRLs was identified. Out of the 281 pesticides that were assessed, 278 were found to be within acceptable levels of exposure for all age groups, and only dieldrin, lindane, and a metabolite of the fungicide propineb, propylene thiourea (PTU), were found to have a significant probability of exceedance of the TRV for several age groups in the upper bound scenario that overestimates exposure. These results indicated the need for more sensitive analyses rather than the high health risk of infants because the high limits of determination (LODs) concerning low TRVs gave overestimated results.

Recently, a study on chlorate in various types of baby foods was published [40], although it was conducted before the reevaluation of the maximum permitted levels of chlorate in baby foods from the European Commission, Commission Regulation 2020/749 [47]. It was found that 10.5% of the samples contained measurable amounts of perchlorate at the concentration range of 3.4–6.5 μg/kg. Five samples (prepared with carrots and potatoes) were detected with chlorate at 40 and 120 μg/kg levels, and one pear sample contained 372.2 μg/kg chlorate. According to Regulation 2020/749, pears’ maximum permitted chlorate level is 50 μg/kg, carrots 150 μg/kg, and potatoes 50 μg/kg. Given these high concentrations, for infants between 4 and 14 months, the average daily dose for pears was equal to or greater than the TDI for chlorate (3 μg/kg body weight/day) (Table 2). 

Serbian infant food (infant juice and purée) was found to have a more significant pesticide burden than imported products, and the risk assessment indicated that there was no considerable health risk for infant health [53]. Therefore, the cumulative risk was calculated using the hazard index and considered all pesticide-active substances as one group, and it was estimated to be negligible. 

Recent dietary risk assessments for young children in studies from China [67,68] are focused on several food commodities (but not specifically infant foods and formulas), and they indicate low levels of exposure to several pesticides and low health risks, although more significant than that for the general population. Ling and co-authors tested food samples (vegetable, fruit, cereal, and seafood) from Taiwan in 2015 for pesticide contamination (neonicotinoids), and the estimated daily intake of children 0–6 years old was found to be within acceptable levels, even for the highly exposed group [54]. 

## 3. Phthalates

Phthalates are widely used plasticizers, mainly polyvinylchloride (PVC). They are found in plastic pipes, medical devices, adhesives, inks and paints, household products such as bath curtains or indoor deodorants, cosmetics, personal care products, plastic food, and water containers [9,69,70]. They are classified into low-molecular-weight phthalates, which are more water-soluble and toxic, and high-molecular-weight phthalates, which are generally considered safer alternatives. The first group includes dimethyl phthalate (DMP) diethyl phthalate (DEP), di-n-butyl phthalate (DnBP), diiso-butyl phthalate (DiBP), and benzyl butyl phthalate (BBP). High-molecular-weight phthalates are di 2-ethylhexyl phthalate (DEHP), diisononyl phthalate (DiNP), diisodecyl phthalate (DiDP), and diisoundecyl phthalate (DiUP). Infant exposure can occur at the beginning of life via medical equipment for hospitalized babies, infant formulas, and breastfeeding [9,69,70,71]. However, prenatal exposure remains unclear, and some animal and human studies indicate the presence of phthalate compounds in fetal plasma, amniotic fluid, meconium, placenta tissue, and urine [72,73,74,75,76,77]. The phthalate compounds detected in biological matrices are phthalate metabolites (hydrolytic or oxidative monoesters), excreted mainly via urine.

The American Academy of Pediatrics and the American Public Health Association are warning about the health effects of phthalates on children as the quantities of food and water that they consume and the inhaled rate are higher relative to their body weight compared to adults, and their not fully developed body is vulnerable and sensitive to the toxicity of phthalates. Although not all phthalates pose the same toxicity, their main effects include endocrine disruption, reproductive toxicity (infertility, diseases, syndromes, hormonal alterations), thyroid hormone alterations, neurodevelopmental effects in infants and children, liver and kidney toxicity, respiratory problems (asthma and allergies), and cancer [9,78,79,80]. 

### 3.1. Current Legislation and Policies for Phthalates

In 2017, the European Union officially designated four phthalates as human endocrine-disrupting chemicals [81]. Namely, BBP, DEHP, DnBP, and DiBP were classified as substances having endocrine-disrupting properties. These compounds already existed on the Candidate List of REACH, although now it is specified that they demonstrate probable severe effects on human health due to their endocrine-disrupting properties. Regarding DiBP, it was initially considered a safer alternative to the more toxic DnBP, and no exposure limit values were established. However, human exposure to DiBP presented a slight increase over the years compared to DnBP exposure, although animal research studies indicated reproductive and developmental effects comparable to DnBP [69,75,82].

The exposure limits for adults (specific exposure limits for infants and children still do not exist) that have been set from the EFSA, expressed as Tolerable Daily Intake (TDI), are 0.01 mg/kg body weight/day for DBP (reproductive effects), 0.5 mg/kg body weight/day for BBP (reproductive effects), 0.05 mg/kg body weight/day for DEHP (reproductive effects), and 0.15 mg/kg body weight/day for both DINP and DIDP due to effects on the liver. In 2019, the EFSA Panel adopted a new approach to the risk assessment of phthalates for use in food contact materials. As DnBP, DEHP, BBP, and DINP have reproductive effects, the EFSA Panel considered it appropriate to establish a group-TDI for these phthalates, which was set at 50 µg/kg body weight/day. For DIDP, the single compound’s TDI remained at 150 µg/kg body weight/day [12,83]. This assessment is for consumers of any age, including adults and vulnerable groups. According to the US EPA, the reference dose for daily exposure to DEHP is 0.02 mg/kg body weight daily.

Especially in the case of phthalates, food packaging may be a potential source of contamination through the migration of chemicals from the packaging into the food. Regarding the detected concentrations of phthalates in infant food and formulas, the Specific Migration Limit (SML) has been set by European Commission to determine the maximum permitted amount of a substance released into food or food simulant from a material. According to Directive 99/39/EC for baby and cereal-based foods, the SML for BBP is 30 mg/kg food simulant, DEHP 1.5 mg/kg food simulant, for DBP 0.3 mg/kg food simulant, and the sum of DINP and DIDP 9 mg/kg food simulant. BBP, DINP, and DIDP are not allowed in single-use material for infants or follow-on formulae and baby food packaging. However, they are permitted in repeated use material and as Technical Support Agents (TSA) in polyolefins with a concentration of up to 0.1% in the final product. DBP and DEHP can be used as a plasticizer in repeatedly used materials contacting non-fatty foods and as TSA in polyolefins with concentrations of up to 0.05% and 0.1% in the final product, respectively [12,83,84].

To reduce the dietary exposure of children and infants to phthalates, the FDA recommends carefully using plastic containers to heat food in microwave ovens. In some products, such as materials in contact with food, the FDA limits using 26 phthalates. For example, BBP should be at a concentration of less than 1% by weight when contained in food-contact polymers. DBP and DEHP can be used alone or in combination with other phthalates with a material content of less than 5% [85,86]. According to Proposition 65, the California Office of Environmental Health Hazard Assessment (OEHHA) listed DINP and DEHP as carcinogenic and DEHP as having reproductive and developmental toxicity.

Finally, according to Health Canada, only DEHP is on the list of toxic substances whose use should be restricted [87], mainly in cosmetics, medical equipment, toys, and childcare products. Other phthalates are considered safe to use, and there is no restriction on the use of phthalates in baby products and food.

### 3.2. Infant Dietary Risk Assessments for Phthalates

The exposure of young children (12–35 months) to phthalates transferred from food packaging to cereal-based food material was investigated by Garcia-Ibarra and co-authors [55] in Spain. The phthalate concentrations in foods were within acceptable levels (0.118 mg/kg for DEP, 0.102 mg/kg for DEHP), and the dietary exposures to DEHP (0.395 μg/kg body weight/day), DEP (0.458 μg/kg body weight/day), and DiBP (0.0864 μg/kg body weight/day) were below the TDI. A total diet study (TDS) was conducted in France by Sirot et al. [56] to evaluate the health risks of infants under three years of age related to the chemicals in food. Phthalates in food from food contact materials and other substances, such as bisphenol A (BPA), bisphenol A diglycidyl ether (BADGE), its derivatives, and some ink photoinitiators, were examined. Generally, phthalates were detected in 0–35% of the samples (DiDP, DBP, BBP, DEHP, and DiNP). The exposure levels of infants to phthalates were within acceptable levels (Table 2). 

Besides monitoring food commodities to estimate dietary exposure, biomonitoring is also considered a reliable alternative method through which levels in urine are converted to daily intake, and a risk assessment can be conducted [75,88]. Following this method, Frederiksen et al. [88] measured fifteen phthalate metabolites in urine from infants during two periods: while exclusively breastfeeding and when they were on mixed diets. The surprising finding was that the infants were exposed to almost all of the examined phthalates. The daily intakes were comparable for many compounds regardless of feeding status, including compounds regulated for years. Notably, the risk assessment analysis indicated that the exposure of infants and their parents sometimes exceeded the safety level for anti-androgenic effects. The results from the specific study show that infants are exposed to phthalates not only through diet, implying the existence of additional sources that are added to the total exposure. Even low levels in infant food and formulas can become dangerous for health if other sources of exposure co-exist and act additively in the infant’s body [89]. 

## 4. Parabens

Parabens are esters of p-hydroxybenzoic acid with an alkyl or benzyl group. They have been widely used as preservatives in foods, cosmetics, personal care products, and medicines for over 90 years due to their anti-microbial and anti-fungal properties [90,91,92]. Parabens are classified as “short-chain” parabens, which are methyl paraben (MeP) and ethyl paraben (EtP), and “long-chain” parabens, which include propyl paraben (PrP), isopropyl paraben (i-PrP), butyl paraben (BuP), isobutyl paraben (iBuP), and benzyl paraben (BzP). Except for their industrial origin, some plants and bacteria can naturally synthesize parabens [90,92,93]. Parabens have been identified as endocrine disruptors that affect androgens, estrogens, progesterone, glucocorticosteroids, aryl hydrocarbon, peroxisome proliferator-activated receptors and the activity of hormones and enzymes that are involved in the metabolism of endogenous hormones and the production of steroids. Breast cancer, obesity, and allergies are some of their potential health effects [91,92,94]. The possible impact on the reproductive system, the hormonal response, and the possible carcinogenic effect have led to the production of paraben-free products labeled as “Paraben Free” [92].

The main routes of exposure are dermal absorption of personal care products and ingesting pharmaceuticals and foodstuffs. In the body, they are metabolized to p-hydroxybenzoic acid, a non-specific metabolite. Parent compounds or their conjugates have been detected in human matrices (blood, plasma, adipose tissue, hair), but they are rapidly excreted in urine after exposure [16,91,95,96,97]. In addition, parabens have been detected in human milk [98,99] and amniotic fluid [100], and studies have reported dietary exposure in infants and young children [57,58], indicating fetal and early-life exposure to the compounds. 

### 4.1. Current Legislation and Policies for Parabens

Based on the EFSA’s risk assessment of consuming foods containing parabens, the Acceptable Daily Intake (ADI) was set at 10 mg/kg body weight for methyl and ethyl parabens and their salts. At the same time, no other limitations were reported for the different parabens. The committee did not recommend ADI for propyl paraben due to the lack of data on adverse effects [101]. Later, the EMA (European Medicines Agency) determined the ADI of propyl paraben at 1.25 mg/kg body weight. The presence of parabens due to the use of veterinary products is expected to be very low in industrially processed foods, so has not been set any maximum residue limit (MRL) [102,103].

Parabens must be mentioned on food labels used as preservatives with the E symbol, for example, methyl paraben as E218 and propyl paraben as E216. The Joint Food and Agriculture Organization of the United Nations (FAO) and the Committee of Experts on Food Additives (JECFA) initially set an ADI of up to 10 mg/kg body weight for methyl, ethyl, and propyl paraben in total. However, the WHO later withdrew propyl paraben from the ADI due to higher cytotoxicity than expected [92]. In the United States, the estimated daily intakes of parabens have been estimated to be 940 ng/kg body weight for infants and 879 ng/kg body weight for children between 1 and 6 years, which are less than the ADI [58].

Parabens have no restrictions in Canada because those naturally present are considered safe, but the limit for daily intake is 10 mg/kg body weight. As additives, their presence is considered harmless for human organisms [104], and they are allowed only in certain parts of the food product, such as topping and/or filling.

### 4.2. Infant Dietary Risk Assessments for Parabens

Many studies focus on maternal exposure to parabens during pregnancy or lactation, mainly through contaminated personal care products. Evaluating the intake of children and infants is highly interesting, even if the available studies are fewer. One study conducted in children aged less than three years old in France showed that dietary exposure to parabens (breastfed infants were excluded) was within acceptable levels [57]. The study suggested that for all examined additives, the exposure increased with age, reaching the highest exposure levels in the age group 13–36 months (mean for parabens 0.35 mg/kg body weight/day, median 0.18 mg/kg body weight/day). Another work was carried out by Liao and co-authors [58], who measured the concentrations of five parabens (benzyl, butyl, methyl, ethyl, and propyl parabens) in foodstuffs including beverages, dairy products, fats and oils, fish and shellfish, grains, meat, fruits, and vegetables. Although it was not focused on infant formulas and baby food, they examined foods that are included in infants’, toddlers’, and children’s diets, and thus they estimated the daily intakes of total parabens for each group at 940 ng/kg body weight/day, 879 ng/kg body weight/day and 470 ng/kg body weight/day, respectively, while for adults it was 307 ng/kg body weight/day (Table 2). Since data in the literature are limited, the results from this study are of great importance. 

## 5. Bisphenols

BPA (BPA, 4,4′-dihydroxy-2,2-diphenylpropane) and its analogs (such as Bisphenols F, S, B, and E) are widely used in industry in the production of polycarbonates, epoxy, and polyester resins, plastic products such as bottles, food packaging, food contact materials, medical equipment, and thermographic and pressure-sensitive paper [12,105,106]. The main routes of exposure to BPs are ingestion and dermal absorption. BPA and its structural analogs have been associated with dysfunctions in the cardiovascular, immune, respiratory, neurological, and endocrine systems, inducing developmental and reproductive problems. Concerning the female reproductive system, BPA has been associated with menstrual and pregnancy disorders, while in males, it has been associated with sexual dysfunction and abnormal sperm parameters [107,108,109]. BPS and BPF present similar toxicity profiles to BPA. Several studies suggested that exposure to BPs, especially during early development, may lead to embryogenesis, placentation, fetal-placental growth, and immune disorders, such as allergies and alterations in the gut microbiome. Obesity, cancer, heart disease, and diabetes are potential health effects of exposure to BPs [6,110,111,112,113]. Bisphenol A has been banned or restricted preemptively in many countries due to increasing concerns about its health effects. Analogs, such as Bisphenols F, S, B, and E, have been replaced in the production of consumer products [114]. 

### 5.1. Current Legislation and Policies for Bisphenols

In Europe, according to the EFSA, Bisphenol A has been classified as an Endocrine Disrupting Chemical, and its analogs have been classified as toxic. The European Commission banned the use of BPA in the baby bottle industry in 2011. The TDI reported by the EFSA is 4.0 μg/kg body weight/day. BPS, one of the BPA analogs, has also been restricted by the EFSA for its use in food contact materials with an SML of 0.05 mg/kg food [11,115]. According to the EFSA (2006) [116], the exposure of breastfeeding infants to BPA is lower (0.2 μg/kg body weight/day) than in 3-month-old infants who are consuming milk from plastic bottles, and it is estimated to be 4 μg/kg body weight/day for normal levels of migration and 11 μg/kg body weight/day for high levels of migration.

In the United States, infant exposure to BPA has been restricted since the FDA banned its use in polycarbonate resins for baby bottles and baby cups, epoxy resins in coatings for infant formula packaging, and metal packaging, which are labeled as “BPA-free”. NOAEL for the general population was determined at 5 mg/kg body weight per day, and for children under two years old, the estimated daily intake (EDI) was 1.1 μg/kg body weight. Following the above ban, BPA consumption is considered safe regarding the allowed uses in food packaging and containers [107,117]. 

In Canada, to protect infants and young children from exposure to BPA, it has been proposed that BPA be used in minimum concentrations in food packaging for newborns and infants (as low as reasonably achievable, ALARA), especially in infant formulas. At the same time, the manufacture, importation, and sale of BPA polycarbonate baby bottles are prohibited [118,119]. 

### 5.2. Infant Dietary Risk Assessments for Bisphenols

The limited studies regarding the infant health risk from exposure to bisphenols via infant formulas and baby food point out the need for more studies to fill this gap. The scientific interest regarding the safety of bisphenols for infants is not limited to BPA but its analogs as well. Seven bisphenols (BPA, BPAF, BPC, BPE, BPFL, BPS, and BPZ) were examined in Indian infant formula, and the highest mean concentration was for BPA (5.46 ng/g) [59]. The calculated EDI of total BPs in infants below one-year-old age was between 54.33 and 213.36 ng/kg body weight/day, and BPA was the compound with the more significant contribution. The risk assessment indicated that exposure was lower than the EFSA reference value (4 μg/kg body weight/day) and thus considered acceptable.

Since the predominant source of exposure to bisphenols is diet, it is interesting to compare the exposure of infants fed with formula with breastfeeding infants [60,61]. The median intake of BPA via breast milk has been calculated to be 26.8 ng/kg body weight/day for newborns (0–3 months) and 7.0 ng/kg body weight/day for infants (4–12 months), according to a study in China [60]. Another study in China, too, found that BPA was the main bisphenol, followed by BPF [61], whose contribution to total exposure was not negligible, indicating that BPA analogs should receive attention. The upper-bound daily intakes of BPs for infants 0–6 months old were found to be between 0.044 and 1.29 μg/kg body weight/day. A comparison between breast and non-breastfed children was made in one study conducted in France [56]. The exposure levels, in some cases, exceeded the reference value established by the French Agency for Food, Environmental and Occupational Health & Safety (0.083 μg/kg body weight/day). However, the EFSA TDI was not exceeded (Table 2). 

## 6. Dioxins, Furans, and Polychlorinated Biphenyls

Dioxins and furans are polyhalogenated aromatic hydrocarbons consisting of 210 congeners, 75 dioxin congeners, and 135 furan congeners, 17 of which are potentially toxic. They are industrial by-products of combustion and other chemical processes, and the most important source of furans are the emissions of cars with halogenated scavengers that use leaded gasoline [120]. According to Javed and co-authors [121], the occurrence of furans in foods is attributed to the thermal degradation of carbohydrates such as glucose and lactose. Thus, furans can be formed from the heat treatment of baby infant formulas, indicating an emerging food safety problem. It seems that the issue of infant exposure to furans is mainly identified in processed ready-to-eat foods in which higher concentrations of furans have been reported than in infant formulas [122]. 

PCBs (polychlorinated biphenyls) consist of 209 synthetic congeners, and their use has been banned in most developed countries. Natural sources have not been identified; they come from industrial emissions and weathering or incineration of materials containing them. They have been widely used in electrical components as paint additives, coolants, and lubricants. Non-ortho and mono-ortho-chloro-substituted diphenols are referred to as “dioxin-like” due to a similar toxicity mechanism to dioxins [14,65]. 

In recent years, there has been a reduction in the environmental residues of these substances due to the restrictions applied; however, due to their high lipophilicity, they are still detected in the environment and human matrices [9,14,65,123,124]. They interfere with nuclear receptors, mainly the aryl hydrocarbon receptor, and due to their structural similarity with thyroxine (T4), they decrease thyroid hormone levels [7,8,125,126]. Disorders of the reproductive system, abnormalities regarding human puberty [9,127,128,129,130], and the immune system [131,132,133] are also reported in the literature. Organohalogens have been identified as hazardous chemicals for human health, and they have been linked with cancer, developmental disorders, hypertension, asthma, metabolic syndromes, and obesity [134,135]. 

### 6.1. Current Legislation and Policies for Dioxins, Furans, and Polychlorinated Biphenyls 

In Europe, the EFSA reports that dioxins and PCBs have been banned since 1980 in most countries. In 2010, a study was published about the levels of dioxins in various foods and feedings, including 219 baby foods. Overall, in baby food, the levels of dioxins were between 0.05 and 0.11 pg TEQWHO_98_/g, furans between 0.06 and 0.08 pg TEQWHO_98_/g, PCBs between 0.16 and 0.17 pg TEQWHO_98_/g, and dioxin-like PCBs between 0.04 and 0.05 pg TEQWHO_98_/g, all fat-based. The average total amount of the above substances was 0.015 pg TEQWHO_98_/g in total baby food weight [136]. In 2012, a study was published about the concentrations of these substances in commercially available baby and child foods, and the maximum concentration was detected in meat or fish-based foods. However, in most samples, the values were below the limit for dioxins and dioxin-like PCBs, at 0.2 pg WHO_2005_-TEQ/g ready-to-eat food (WHO_2005_ refers to the update of the coefficients by WHO in 2005) [115,123]. Finally, the tolerable weekly intake (TWI) set by the EFSA is 2 pg/kg body weight [116].

As stated by the FDA, an additional source of PCBs in food could be paper packaging, mainly by recyclable material. For baby or infant foods, the residues of PCBs should be below 0.2 ppm [137], but no legislation is aimed at restricting dioxins in baby food.

### 6.2. Infant Dietary Risk Assessments for Dioxins, Furans, and Polychlorinated Biphenyls 

Most recent studies are focused on estimating infant exposure through breastfeeding. Lin and co-authors [62] measured polybrominated dibenzo-p-dioxins and dibenzofurans (PBDD/Fs) in human milk samples and investigated the health risk to breastfed infants. The results indicated that the average estimated dietary intake (EDI) for breastfed infants was 2.0 pg TEQ/ kg body weight/day, which is within acceptable levels according to TDI for TCDD suggested by the WHO (1–4 pg TEQ/kg body weight/day). However, according to the authors, given the high toxicity of PBDD/Fs, the potential health risks of these pollutants for breastfed infants should be of concern. Another similar study [63] conducted in Uganda in 2018 measured POPs in breast milk and found infant EDIs for dioxins that exceeded the reference values from the WHO in most samples. Bruce-Vanderpuije et al. [64] measured polybrominated and mixed halogenated dibenzo-p-dioxins and furans (PBDD/Fs and PXDD/Fs) and dioxin-like polychlorinated biphenyls (DL-PCBs) in 24 human milk samples of mothers from Ghana and the results showed greater infant intake than the recommended standard intake of 1 pg TEQ/kg body weight/day as set by the ATSDR and WHO. The EDI of DL-PCBs in 21 human milk samples was 4.95 pg TEQ/kg body weight/day; contributions from DL-PCBs, PXDD/Fs, and PBDD/Fs resulted in an average estimated daily intake of 6.56 pg TEQ/kg body weight/day (Table 2).

A French study [65] measured PCBs and dioxins in 180 food samples to assess PCB exposure through the whole diet of non-breastfed children from 1 to 36 months old (705 participants). The levels of PCDD/Fs and PCBs in infant food were lower than in everyday food. However, for dioxins and PCBs, the TDI was exceeded for older age groups (it mainly concerned standard milk for the youngest children, ultra-fresh dairy products, and fish). An older study [66] estimated the dietary intake of PCDD/Fs and dioxin-like PCBs in Greece from infant formula levels and food items in the Greek market (2002–2010) and human milk samples. The significant results that were obtained indicated that breastfed infants (0–6 months) had greater TDI (60.3–80.4 TEQ pg/kg body weight) than infants that consumed human milk and formula (31.2–41.6 TEQ pg/kg body weight). For breastfed infants between 6 and 12 months, TDIs were 19.76–24.95 TEQ pg/kg body weight for breastfed, and for infants receiving only formula, the TDIs ranged from 1.60 to 2.24 TEQ pg/kg body weight. 

## 7. Conclusions

Human milk, which is the optimal option for the nutrition of infants, can also be contaminated with several pollutants, but it is hard to control this burden. Moreover, the burden of infant formulas and baby food can be partially controlled since some chemicals may not be fully eliminated but can be kept at the lowest achievable level. Therefore, the results of monitoring and risk assessment studies can evaluate the effectiveness of the applied regulations and policies. Currently, these studies indicate that it is not common to detect high exposure levels of infants to the discussed chemicals via infant formulas, especially in European countries and the USA. However, the cumulative, multicomponent exposure and the sensitivity of the still-developing body cannot be underestimated. Therefore, the adverse effects of low-dose exposure in newborns may become evident over time. 

Regarding the three regulatory agencies discussed in the present study, the European Commission, the US EPA, and Health Canada, different policies and approaches are applied under the same goal: food safety. There has been a particular emphasis on the exposure of infants and newborns as a vulnerable population group to some chemicals. However, exposure limits for infants and children do not exist for others. 

In Europe, pesticide residues in cereal-based and baby foods should not exceed the amount of 0.01 mg/kg food, and new labeling rules have been applied to infant formulas [42]. The EPA evaluates new and existing pesticides to ensure they are used safely for infants, children, and adults, and reevaluates each pesticide every 15 years. Additional safety factors are established for the safety of pesticide use on food to consider the uncertainty in data relative to children [48,49]. In Canada, when setting restrictions and prohibitions on pesticides, special emphasis is placed on the young population, from the embryonic stage to childhood [50,51].

The exposure limits for adults (specific exposure limits for infants and children still do not exist) that have been set from the EFSA, expressed as a group-TDI, for four phthalates was 50 µg/kg body weight/day [12,83]. According to the US EPA, the reference dose for daily exposure to DEHP is 0.02 mg/kg body weight daily. To reduce the dietary exposure of children and infants to phthalates, the FDA recommends carefully using plastic containers to heat food in microwave ovens. In some products, such as materials in contact with food, the FDA limits using 26 phthalates [85,86].

Based on the EFSA risk assessment of consuming foods containing parabens, the ADI was set at 10 mg/kg body weight for methyl and ethyl parabens and their salts; the EMA determined the ADI of propyl paraben at 1.25 mg/kg body weight [101,102]. There are no restrictions for parabens in Canada because those naturally present are considered safe, but the limit for daily intake is 10 mg/kg body weight [104]. Therefore, the Joint Food and Agriculture Organization of the United Nations (FAO) and the Committee of Experts on Food Additives (JECFA) initially set an acceptable daily ADI of up to 10 mg/kg body weight for methyl, ethyl, and propyl paraben in total. 

The European Commission banned using BPA in the baby bottle industry in Europe in 2011. The TDI reported by the EFSA is 4.0 μg/kg body weight/day. BPS, one of the BPA analogs, has also been restricted by the EFSA for its use in food contact materials with an SML of 0.05 mg/kg food [11,115]. In the United States, the FDA banned its use in polycarbonate resins for baby bottles and cups, epoxy resins in coatings for infant formula packaging, and metal packaging labeled as “BPA-free” [107,117]. In Canada, to protect infants and young children from exposure to BPA, it has been proposed that BPA be used in minimum concentrations in food packaging for newborns and infants (as low as reasonably achievable, ALARA), especially in infant formulas [118,119].

The tolerable weekly intake (TWI) set by the EFSA for PCBs is 2 pg/kg body weight [116]. As stated by the FDA, for baby or infant foods, the residues of PCBs should be below 0.2 ppm, but there is no legislation to restrict dioxins in baby food [137].

Recent data on risk assessment indicate that dietary exposure of infants and young children to several types of pollutants should be continuously under investigation and monitoring. Special focus should be given to infant formulas and baby foods for compounds that potentially pose health risks, whether they are subjected to regulation or not. In addition, more studies should be conducted to fill gaps regarding the insufficient number of risk assessment studies, especially multiple and combined exposure in vulnerable groups. Finally, all regulatory agencies should establish threshold limits of exposure specifically for infants and implement practices to remove toxic compounds from young children’s food sources adequately.

## Figures and Tables

**Table 1 foods-12-01697-t001:** Chemical names, usage, and reported effects of specific compounds with potential endocrine action *.

Chemical Names	Usage	Effects
Organophosphate pesticides (OPs)	Agriculture	Neurological effects Respiratory problems Hypospadias Birth effects
Dioxins (PCDDs)	Mainly by-products of industrial practices	Delayed breast development during adolescence Increased possibility of female birth
Polychlorinated biphenyls (PCBs)	Industry	Male infertility Disorders of neuro-behavioral, respiratory, and immune systems in children, Diabetes
Phytoestrogens	Agriculture	Hypospadias
Flame retardants, such as tetrabromobisphenol A (TBBPA), PBDEs, and PBBs	Industry	Cryptorchidism Diabetes Neuro-developmental disorders Potentially carcinogenic Thyroid disruption
Polybrominated biphenyls (PBBs)	Industry	Early onset of menstruation
Diethylstilbestrol (DES)		Male: hypospadias, cryptorchidism, epididymal cysts, and disorders of testicular and sperm function Female: increased risk of breast cancer, vaginal adenosis, oligomenorrhea, genital cancers, and pregnancy disorders
Organochlorine pesticides (DDT and its metabolites)	Industry, agriculture	Possibly carcinogenic Delayed adolescence 4,4′-DDE: anti-androgenic
Perfluorinated chemicals (PFC), such as perfluorooctanoic acid (PFOA) and perfluorooctane sulfonate (PFOS)	Coatings against stains and oils, floor varnishes, and pesticides	Endocrine disruption
Phthalates	Plastic packaging	Disruption of androgenic biosynthesis Variation in the androgenic index Increased LH/testosterone ratio
Bisphenol A (BPA) and analogs	Plastic or coatings in food packaging	Endocrine disruption
UV filters, such as 4-methylbenzylidene-camphor (4-MBC), octyl-methoxycinnamate (OMC), and benzophenone 2 (BP2)	Sunscreens	Endocrine disruption

* Refs. [9,11,12,13,14,15].

**Table 2 foods-12-01697-t002:** Results of risk assessments from studies in literature for several categories of EDCs.

Reference	Compounds	Study Information	Results
Pesticides			
[52]	281 pesticides for risk assessment (OCPs, neonicotinoids, carbamates, OPs)	French study, 2011–2012, 219 baby foods, 705 infants and young children (<3 years old)	278 pesticides with acceptable risk, the high LODs gave overestimated results for three pesticides
[40]	Pesticides (chlorates and perchlorates)	105 baby food samples, infants (4–24 months)	Daily dose of chlorate for high concentrations in vegetables: 1.2–2.1 μg/kg BW/day, and pears: 2.5–4.3 μg/kg BW/day (EFSA TDI: 3μg/kg BW/day)
[53]	69 pesticides (OCPs, OPs, neonicotinoids, carbamates, triazoles, etc.)	54 Serbian infant food samples (infant juice and purée)	Cumulative risk via hazard index was estimated to be negligible
[54]	Pesticides (7 neonicotinoids)	128 food samples from Taiwan	Residues in the diets of preschool children did not exceed ADI
		**Phthalates**	
[55]	DEP, DIBP, DBP, BBP, DEHP	Young children (12–35 months) in Spain, cereal-based food samples	Mean dietary exposure 1.01 μg/kg BW/day below TDI
[56]	BBP, DBP, DEHP, DIDP, DINP	Evaluated the health risks of infants (<3 years old) related to phthalates in food coming from food contact materials	The exposure levels of infants to phthalates were within acceptable levels
		**Parabens**	
[57]	6 parabens (E214–E219)	Children (<3 years old) in France (breastfed infants were excluded)	The highest exposure levels in the age group 13–36 months (mean 0.35 mg/kg BW/day, median 0.18 mg/kg BW/day). Dietary exposure to parabens was within acceptable levels
[58]	5 parabens (benzyl, butyl, methyl, ethyl, propyl parabens)	Foods that are included in infants’, toddlers’, and children’s diets	Estimated daily intakes of total parabens for infants, toddlers, and children, respectively: 940 ng/kg BW/day, 879 ng/kg BW/day, and 470 ng/kg BW/day
		**Bisphenols**	
[59]	7 bisphenols (BPA, BPAF, BPC, BPE, BPFL, BPS, BPZ)	Infants (<1 year old), Indian infant formula	EDI of total BPs ranged from 54.3 to 213.4 ng/kg BW/day. Risk assessment indicated that exposure was considered acceptable
[60]	Bisphenol-A	Breastfeeding infants	Median intake of BPA 26.8 ng/kg BW/day for newborns (0–3 months) and 7.0 ng/kg BW/day for infants (4–12 months)
[61]	12 bisphenols	181 breastmilk samples collected from China in 2014	The upper-bound daily intakes of BPs for infants 0–6 months old were found between 0.044 and 1.29 μg/kg BW/day
[56]	Bisphenol-A	Comparison between breastfed and non-breastfed children in France	The TDI of the EFSA was never exceeded
		**POPs**	
[62]	PBDD/Fs	Human milk samples and health risks to breastfed infants in China	The average EDI was 2.0 pg TEQ/kg BW/day (0.13–13 pg TEQ/kg BW/day), within the range of TDI by the WHO
[63]	PCBs, PCDD/Fs	30 human milk samples from Uganda in 2018	Potential health risks to nursing infants associated with consumption of breastmilk
[64]	PBDD/F, PXDD/F, and dlPCBs	24 human milk samples in Ghana	Greater infant intake than the recommended standard intake of 1 pg TEQ/kg BW/day as set by the ATSDR and WHO
[65]	PCBs, PCDD/Fs	180 food samples, dietary exposure for 705 children under 3 years of age	For dioxins and NDL-PCBs, the TDI was exceeded for some age groups, in particular for older ones
[66]	PCBs, PCDD/Fs	Levels in infant formulas, food items in the Greek market (2002–2010), and human milk samples	Breastfed infants (0–6 months) had greater TDI (60.3–80.4 TEQ pg/kg BW) than infants that consumed human milk and formula (31.2–41.6 TEQ pg/kg BW)

## Data Availability

The data presented in this study are available on request from the corresponding authors.

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
