# Peer review of "Endocrine-Disrupting Chemicals and Persistent Organic Pollutants in Infant Formulas and Baby Food: Legislation and Risk Assessments"

_foods, 2023, doi:10.3390/foods12081697_

Round 1

Reviewer 1 Report

The topic chosen is of great interest from a toxicological and food safety perspective. Risk managers and risk evaluators need more data to fill the actual gaps of knowledge still existing. Research must be focused in those gaps and needs. Climate change is affecting the occurrence of different toxicants and risks. The food and Pharmaceutical industry should commit with minimizing the exposure to plastics and its additives and other inmune toxicants.

The whole text should be enriched with a bigger number of references as the paper has been submitted as a review

Table 1 needs to be based on a more recent review.

What about inserting EFSA as a reference?

Why did authors just insert a table to review the pesticides previous studies and not the other compounds discussed?. Every family of endocrine disruptors and toxics should have a table presenting previous data.

The whole conclusions section must be rewritten since it is too short and poor. More concrete actions must be recommended to industry, regulators, market, consumers and academia (researchers).

Author Response

We thank the reviewers for their valuable comments and their contribution for the enrichment and improvement of the manuscript. Below are our responses to their comments.

The topic chosen is of great interest from a toxicological and food safety perspective. Risk managers and risk evaluators need more data to fill the actual gaps of knowledge still existing. Research must be focused in those gaps and needs. Climate change is affecting the occurrence of different toxicants and risks. The food and Pharmaceutical industry should commit with minimizing the exposure to plastics and its additives and other inmune toxicants.

The whole text should be enriched with a bigger number of references as the paper has been submitted as a review

More references were added in the reference list of the revised manuscript according to the suggestion of the reviewer.  

Table 1 needs to be based on a more recent review.

What about inserting EFSA as a reference?

The information that was presented in table 1 in the initial version was recent. However, according to the reviewer’s comment EFSA and other recent studies were also cited: EFSA, 2015; EFSA, 2019b; Laws et al., 2021; Li et al, 2006; Stohs, 2014; WHO, 2012. The citations were removed from the caption and placed below the Table.

Why did authors just insert a table to review the pesticides previous studies and not the other compounds discussed?. Every family of endocrine disruptors and toxics should have a table presenting previous data.

Table 2 (Caption: Results of risk assessments from studies in literature for several categories of EDCs.) presents the results from all studies for all chemicals that are discussed in the risk assessment part. After revision, a more distinct partition of the table was done. The Table was cited in paragraphs 2.2 (risk assessment for pesticides) and 6.2 (risk assessment for POPs). After revision, the table is cited also in paragraphs 3.2, 4.2 and 5.2.

The whole conclusions section must be rewritten since it is too short and poor. More concrete actions must be recommended to industry, regulators, market, consumers and academia (researchers).

The conclusions were revised according to both reviewers’ comments.

We thank the reviewers for their valuable comments and their contribution for the enrichment and improvement of the manuscript. Below are our responses to their comments.

The topic chosen is of great interest from a toxicological and food safety perspective. Risk managers and risk evaluators need more data to fill the actual gaps of knowledge still existing. Research must be focused in those gaps and needs. Climate change is affecting the occurrence of different toxicants and risks. The food and Pharmaceutical industry should commit with minimizing the exposure to plastics and its additives and other inmune toxicants.

The whole text should be enriched with a bigger number of references as the paper has been submitted as a review

More references were added in the reference list of the revised manuscript according to the suggestion of the reviewer.  

Table 1 needs to be based on a more recent review.

What about inserting EFSA as a reference?

The information that was presented in table 1 in the initial version was recent. However, according to the reviewer’s comment EFSA and other recent studies were also cited: EFSA, 2015; EFSA, 2019b; Laws et al., 2021; Li et al, 2006; Stohs, 2014; WHO, 2012. The citations were removed from the caption and placed below the Table.

Why did authors just insert a table to review the pesticides previous studies and not the other compounds discussed?. Every family of endocrine disruptors and toxics should have a table presenting previous data.

Table 2 (Caption: Results of risk assessments from studies in literature for several categories of EDCs.) presents the results from all studies for all chemicals that are discussed in the risk assessment part. After revision, a more distinct partition of the table was done. The Table was cited in paragraphs 2.2 (risk assessment for pesticides) and 6.2 (risk assessment for POPs). After revision, the table is cited also in paragraphs 3.2, 4.2 and 5.2.

The whole conclusions section must be rewritten since it is too short and poor. More concrete actions must be recommended to industry, regulators, market, consumers and academia (researchers).

The conclusions were revised according to both reviewers’ comments.

Reviewer 2 Report

The review is well-written and includes recent data on endocrine disrupters (EDCs) in infant formula suggesting it is an accurate reflection of current exposure.  I have the following comments regarding the manuscript:

1. The conclusion needs to be expanded.  This should be the section where everything is tied together and general statements or conclusions are reached.  Two sentences is not an adequate discussion for this topic.  I recommend a more thorough summary of the data and describing the specific data gaps.  Additionally, this manuscript has the unique perspective of including three major regulatory agencies and their approaches to assessing risk for these chemicals.  The conclusion section is a great opportunity to discuss the similarity and differences in their approach and whether this changes the conclusions regarding safety.

2. There is a review of EDCs in baby formula published by Yesildemir and Yasemin (2021).  The referenced article appears to contain studies that were not included in this review.  Please access the article and determine if any of the referenced studies can add to this manuscript.(http://sjafs.selcuk.edu.tr/sjafs/article/view/1249).

3. Page 4 states "the present study aims...".  This is a review article and I suggest replacing the word "study" with "review" or "analysis".

4. The reference on the bottom of page 17 needs to be corrected.  It is currently a link to the document.

Author Response

We thank the reviewers for their valuable comments and their contribution for the enrichment and improvement of the manuscript. Below are our responses to their comments.

The review is well-written and includes recent data on endocrine disrupters (EDCs) in infant formula suggesting it is an accurate reflection of current exposure.  I have the following comments regarding the manuscript:

  1. The conclusion needs to be expanded.  This should be the section where everything is tied together and general statements or conclusions are reached.  Two sentences is not an adequate discussion for this topic.  I recommend a more thorough summary of the data and describing the specific data gaps.  Additionally, this manuscript has the unique perspective of including three major regulatory agencies and their approaches to assessing risk for these chemicals.  The conclusion section is a great opportunity to discuss the similarity and differences in their approach and whether this changes the conclusions regarding safety.

We thank the reviewer for the apt comment. The conclusions were revised according to both reviewers’ suggestions.

  1. There is a review of EDCs in baby formula published by Yesildemir and Yasemin (2021).  The referenced article appears to contain studies that were not included in this review.  Please access the article and determine if any of the referenced studies can add to this manuscript.(http://sjafs.selcuk.edu.tr/sjafs/article/view/1249).

We thank the reviewer for the recommendation. New information has been added from the specific article in paragraph 6: “According to Javed and co-authors (2021), the occurrence of furans in foods is attributed to the thermal degradation of carbohydrates such as glucose and lactose. Thus, furans can be formed from the heat treatment of baby infant formulas indicating the emerging food safety problem.It seems that the problem of infant exposure to furans is mainly identified in processed ready-to-eat foods in which higher concentrations of furans have been reported than infant formulas (EFSA, 2011). ” …. “Organohalogens have been identified as hazardous chemicals for human health and they have been linked with types of cancer, developmental disorders, hypertension, asthma, metabolic syndromes and obesity (Antignac et al., 2016; Yesildemir and Akdevelioglu, 2021).”

  1. Page 4 states "the present study aims...".  This is a review article and I suggest replacing the word "study" with "review" or "analysis".

The word “study” was replaced with “review”.

  1. The reference on the bottom of page 17 needs to be corrected.  It is currently a link to the document.

The reference in text was replaced with EPA Web Archives and it was added in the reference list at the end.

Round 2

Reviewer 1 Report

Thanks for answering the round 1 review.

Abstract: why saying "There are differences in legislation in Europe, the USA and Canada" and not just worldwide as lines 6 and 14 of introduction mention?. Otherwise, why these 3 palces and not others? Line 235 reviews a risk assessment from China, line 240 from Taiwan

Reorder key words trying to put together the similar concepts: baby food; infant formulas;  endocrine disruptors;  legislation; persistent organic pollutants; pesticides; plasticizers; risk assessment; exposure limits

Table 1: are Dioxins used by industry or a side product of industry?.

Table 1: sometimes there are Caps letters to start a stentence and others not. Please follow the same criteria

Line 428 mentiosn Pub Med: authors should refer the date of that reserach on line as well as the filters to reach that conclusion

Line 627: highexposure must be separarated in 2 words

Conclusion section needs a final sentence to resume all compounds. Something suggesting future steps.

Author Response

We thank again the reviewer for the valuable comments.

  1. Abstract: why saying "There are differences in legislation in Europe, the USA and Canada" and not just worldwide as lines 6 and 14 of introduction mention?. Otherwise, why these 3 palces and not others? Line 235 reviews a risk assessment from China, line 240 from Taiwan

The sentence “There are differences in legislation in Europe, the USA and Canada” in the Abstract was revised according to the reviewer’s comment “Legislation differs around the world, although there are common policies and strategies for the protection of infants which is a vulnerable group.”

  1. Reorder key words trying to put together the similar concepts: baby food; infant formulas;  endocrine disruptors;  legislation; persistent organic pollutants; pesticides; plasticizers; risk assessment; exposure limits

Keywords were placed alphabetically but they were reordered according to the reviewer’s comment.

  1. Table 1: are Dioxins used by industry or a side product of industry?.

Table 1 was revised. Dioxins are mainly byproducts of industrial practices.

  1. Table 1: sometimes there are Caps letters to start a stentence and others not. Please follow the same criteria

Table 1 was revised.

  1. Line 428 mentiosn Pub Med: authors should refer the date of that reserach on line as well as the filters to reach that conclusion

This search was done 1 to 2 years ago when the manuscript was under development. The advanced search contained the keywords parabens, infant and diet. This paragraph was revised.

  1. Line 627: highexposure must be separarated in 2 words

It seems that something is happening with the word file as new missing spaces are appearing when downloading a revised version.

  1. Conclusion section needs a final sentence to resume all compounds. Something suggesting future steps.

Conclusions were revised.

Reviewer 2 Report

All of my comments have been addressed.  The conclusion is much improved and adequately summarizes the data.

Author Response

We thank the reviewer for the valuable contribution.